# Fibroblast Memory in Development, Homeostasis and Disease

**DOI:** 10.3390/cells10112840

**Published:** 2021-10-22

**Authors:** Thomas Kirk, Abubkr Ahmed, Emanuel Rognoni

**Affiliations:** Centre for Endocrinology, William Harvey Research Institute, Barts and the London School of Medicine and Dentistry, Queen Mary University of London, London EC1M 6BQ, UK; t.b.kirk@qmul.ac.uk (T.K.); abubkr.ahmed@qmul.ac.uk (A.A.)

**Keywords:** biological memory, fibroblasts, wound healing, fibrosis, cancer, inflammation, metabolism, positional identity, epigenetic modification, mechanical stress, cell fate

## Abstract

Fibroblasts are the major cell population in the connective tissue of most organs, where they are essential for their structural integrity. They are best known for their role in remodelling the extracellular matrix, however more recently they have been recognised as a functionally highly diverse cell population that constantly responds and adapts to their environment. Biological memory is the process of a sustained altered cellular state and functions in response to a transient or persistent environmental stimulus. While it is well established that fibroblasts retain a memory of their anatomical location, how other environmental stimuli influence fibroblast behaviour and function is less clear. The ability of fibroblasts to respond and memorise different environmental stimuli is essential for tissue development and homeostasis and may become dysregulated in chronic disease conditions such as fibrosis and cancer. Here we summarise the four emerging key areas of fibroblast adaptation: positional, mechanical, inflammatory, and metabolic memory and highlight the underlying mechanisms and their implications in tissue homeostasis and disease.

## 1. Introduction

To respond appropriately to a dynamic external environment, cells require memory. Biological memory defines the process of transcriptional and epigenetic priming of a cell for a specific cellular state or fate in response to a transient or permanent extrinsic stimulus. These environmental stimuli may include changes in or exposure to stiffness, tissue integrity (UV damage), inflammation, extracellular signalling, or cell-cell/cell-matrix interactions. Memory in cells is maintained through a variety of mechanisms including direct DNA and chromatin modifications, cytoplasmic components, and extracellular signalling and contacts (Figure 1). Cellular memory is best known in the case of the adaptive immune system, in which DNA recombination and clonal selection in response to antigens lead to the production of a population of cells with a ‘memory’ for a specific antigen. Immune memory was long thought to be limited to cells of the adaptive immune system. However, it is now understood that innate immune cells also possess the ability to be primed by an initial exposure and present an enhanced response upon subsequent exposure to infection through epigenetic reprogramming, a process termed ‘trained immunity’ [1]. The skin forms part of the innate immune system, functioning as a barrier to infection and when this fails, it is the site for the early response to pathogens. Non-haematopoietic cells of the skin such as epithelial stem cells exhibit memory of inflammation through maintained expression of AIM2 [2]. Fibroblasts are emerging as important inflammatory mediators in homeostasis, wounding, and disease; it is therefore conceivable that fibroblasts also exhibit immune memory.

Fibroblasts are the most common cells of connective tissues, but are a poorly defined population, with limited specific, universal markers [3]. They have classically been defined as structural cells that specialise in the deposition and remodelling of the extracellular matrix (ECM) and are essential for the maintenance of the tissue integrity. While in homeostasis most fibroblasts quiesce, they become rapidly activated upon tissue damage, start proliferating and migrate to the site of tissue damage, in which they deposit and remodel large amounts of ECM. Today, fibroblasts are increasingly recognised for their contribution to immune surveillance and inflammation, blood vessel function, cancer progression, and maintenance of tissue specific structures (e.g., hair follicles) and stem cell niches (e.g., bone marrow, synovium). Recent advances in single cell transcriptomics have revealed that fibroblasts display significant functional heterogeneity, with fibroblast functions varying by their anatomical location and microenvironment [4]. There is ongoing work to generate a single cell atlas of fibroblast diversity across the whole organism in mice and humans in order to understand fibroblast development, heterogeneity, and disease [5].

Currently, the molecular basis and significance of fibroblast heterogeneity within a tissue and across multiple different organs is largely unknown. Intriguingly, perturbations of the microenvironment or transplantation to a different anatomical location influences fibroblast behaviour, but some fibroblast features persist. Thus, fibroblasts seem to have the capability to adapt and respond to a new stimulus as well as to retain a memory of their past environmental stimuli (e.g., tissue damage, inflammation). Beside their positional identity, it is emerging that fibroblasts in different organs are able to memorise changes to their mechanical (tissue stiffness), inflammatory and metabolic environment. These memories may be short-term, such as in the case of mechanical stimuli or long-term in the case of positional memory, which is important for maintaining homeostasis, directing regeneration, and controlling inflammation. Here we aim to dissect the underlying molecular mechanisms of four major types of fibroblast memory: mechanical, positional, immune, and metabolic (Figure 2) and discuss their involvement in tissue homeostasis and disease.

## 2. Positional Memory

Anatomical location is an important organising principle for a diverse number of cell types in multicellular organisms [6]. Fibroblasts are the major structural cells in almost every organ which differentiate into tissue specific subpopulations during morphogenesis. The demands on connective tissue (e.g., physical and mechanical challenges, environmental stress and ageing, permeability and elasticity, cellular and molecular composition, etc.) are highly heterogenous across body sites and within organs such as the skin, joints, lung or heart. Although we don’t yet fully understand the extent of anatomical variation of connective tissue functions, maybe the positional identity of the fibroblast helps in establishing and maintaining specialist features during development, homeostasis and repair.

In the hallmark study by Chang et al., comparison of foetal and adult fibroblasts from 10 different anatomical locations has revealed distinct gene expression patterns between different organs but also within a tissue along the developmental body axes reflecting their embryonic origins [7]. Approximately 8% of all genes transcribed in fibroblasts are differentially expressed in a site-specific manner and are involved in regulation of ECM synthesis, lipid metabolism, and cell signalling pathways such as TGF-β, Wnt and GPCR controlling proliferation, cell migration, and differentiation. For example, while both foetal lung and skin fibroblasts express high levels of type IV collagen, a central component of the basement membrane, only dermal fibroblasts synthesise type I and V collagen, which is essential for the tensile strength of adult skin dermis. Similarly, differentiation factors such as FOXF1 that are essential for the lung branching morphogenesis are restricted to foetal lung fibroblasts. Intriguingly, the site-specific transcriptional differences can be maintained long-term in vitro and are not influenced by different culture conditions (asynchronous cell growth or in serum-free media conditions), establishing the concept of positional memory in fibroblasts. A subsequent, largescale study of primary fibroblasts utilising adult human tissue samples extracted across 43 unique body sites discovered differences that related to the three anatomical axes: anterior-posterior, proximal-distal, and dermal versus non-dermal [8]. Analysis of the 317 genes that were enriched in fibroblast samples across these different sites revealed several HOX genes, which are known master regulators of positional identity during body morphogenesis. Indeed, clustering of fibroblasts based on 51 key homeodomain transcription factors was able to map their respective anatomical location. While expression of the *HOXB* gene is limited to the trunk and non-dermal samples, *HOXD4* and *HOXD8* are exclusively expressed in the trunk and proximal leg samples and *HOXA13* is only present in adult fibroblasts extracted from distal sites. Functionally, continuous HOX gene activity appears to be vital in adult cells for enabling persistent expression of genes relevant to their positional identity within the tissue. For example, *HOXA13* activity in adult fibroblasts maintains the expression of WNT5A and epidermal keratin 9, which is essential for their distal-specific transcriptional program [9], highlighting their importance for tissue development and homeostasis. As in dermal or lung fibroblasts, positional HOX gene signatures are sufficient to discriminate synovial fibroblasts in the joints from different body sites. A transcriptomic screening of synovial fibroblasts from different anatomical sites and patients with different clinical pathologies revealed that fibroblasts clustered according to anatomical location rather than disease type or progression [10]. The synovial fibroblast samples could be assigned to the original joint location by clustering the transcripts from HOX loci, emphasising the importance of HOX gene activity for their positional identity. Here, *HOXA* and *HOXD* gene transcripts define positional identity of distal synovial fibroblasts of the hand joints, whereas shoulder-derived synovial fibroblasts express a combination of *HOXA*, *HOXB* and *HOXD* transcripts; *HOXC* locus transcripts also distinguish knee from upper extremity synovial fibroblasts. In addition, a large-scale RNA-seq analysis of primary human fibroblasts from healthy cadavers confirmed that fibroblast heterogeneity clusters among different anatomical locations as opposed to the donors, pointing to a highly conserved fibroblast tissue diversity [11].

Intriguingly, site-specific HOX gene expression patterns in adult fibroblasts from different anatomical sites persist over multiple passages in culture and are not influenced by co-culture or conditioned media of fibroblasts from different locations/origins [7,8]. In line, cell transplantation experiments between different anatomical sites revealed that transplanted fibroblasts largely maintain their cellular identity and scarring behaviour of their previous positional location in vivo [12,13]. In contrast, treatment with Trichostatin A, a histone deacetylase inhibitor, was sufficient to significantly perturb HOX gene activity in adult fibroblasts, indicating that their expression and thus their positional identity is maintained intrinsically by distinct chromatin modifications including histone acetylation and methylation [14,15,16]. DNA methylations in the *HOXC* locus were shown to discriminate knee from hand and shoulder synovial fibroblasts [10]. Alongside DNA methylations and histone modifications, joint-specific HOX gene expression was suggested to be regulated by bromodomain and extra-terminal reader (BET) proteins, which are acetylated histone-binding proteins that remain associated during mitosis and potentially influence transcriptional memory during cell division. In addition to canonical HOX genes, HOX non-coding RNAs (ncRNA) vary significantly in expression from different anatomical sites [17]. ncRNAs are enriched for specific DNA sequence motifs based upon their anatomical location that may represent binding sites for DNA or RNA regulatory factors influencing the epigenetic gene regulation. The ncRNA HOTAIR, residing in the *HOXC* locus, for example, was shown to repress the transcription of the *HOXD* locus by interacting with the Polycomb Repressive Complex 2 (PRC2) [17].

Thus, highly distinct and epigenetically maintained patterns of HOX gene expression promote positional memory in fibroblasts as well as their tissue and anatomical site-specific transcriptional signatures and, potentially, their functions (Figure 2a). While the positional memory of different anatomical sites is very stable, within the tissue architecture the positional identity of fibroblasts is much more plastic and strongly influenced by the surrounding microenvironment and tissue state (e.g., development, disease, ageing). Skin fibroblasts of the upper (papillary) and lower (reticular) dermis lose their specific marker expression and cellular identity with age [18,19,20]. This loss can be reversed through transgenic induction of epidermal Wnt signalling (expression of stabilised β-catenin in epidermal keratinocytes), leading to an expansion of the papillary fibroblast population in the upper dermis [21]. Within the skin the persistence of positional identity differs between dermal fibroblast subpopulations and seems to correlate with their differentiation state and plasticity. Papillary and reticular fibroblast identity are quickly lost in conventional tissue culture [22,23], whereas dermal papilla fibroblasts or adipocytes identity are maintained over several passages in vitro [24,25]. During wound healing, fibroblasts of the upper and lower dermis randomly redistribute within the dermis and promote the regeneration of hair follicle associated fibroblast subpopulations, pericytes or adipocytes, respectively [26,27]. Notably, how these fibroblast lineages contribute to the regeneration of blood vessel associated pericytes is dependent on the location of the regenerating blood vessel within the wound bed [27]. While specialised fibroblast subpopulations of the dermal papilla, dermal sheath and arrector pili muscle are unable to participate in wound repair [28,29], adipocytes of the dermal white adipose tissue (DWAT) were shown to differentiate into wound bed myofibroblasts during the early wound repair phase in response to TGF-β signalling [30,31]. Interestingly, these activated (myo)fibroblasts have the potential to convert to quiescent adipocytes during the later wound resolution phase [32] or adipocytes-derived myofibroblasts maintain a positional memory and convert back to their original cellular fate

Also, in other tissues such as the lung, kidney, heart or intestine, multiple functionally and spatially distinct fibroblast subpopulations have been discovered by recent advances in single cell RNA-seq and lineage tracing technologies (reviewed in [4]); however, the underlying intrinsic and extrinsic mechanisms regulating their plasticity and positional identity have only begun to be explored (Figure 2a). Interplay between anatomical and tissue-specific positional identity warrant further investigation and may be influenced by different environmental stimuli including tissue stiffness or extracellular signals.

## 3. Mechanical Memory

In their role as structural cells of tissues, fibroblasts must sense the mechanical environment which in turn instructs their cell behaviour and fate during development, homeostasis and disease. Some of these changes are able to persist for longer time periods after removal of the mechanical stimulus, establishing the concept of mechanical memory [33]. Synthesis of ECM components are coupled to mechanical sensing to maintain homeostasis and establish the tissue architecture. In the skin, for example, the dermal maturation is governed by a coordinated switch in fibroblast behaviour from highly proliferative in embryonic development to quiescence, and high ECM deposition/remodelling postnatally that is maintained by the surrounding ECM network [34]. Upon organ injury, disruption of the mechanical tissue integrity results in enhanced mechanical stress, differentiation to myofibroblasts and increased ECM synthesis/remodelling. The increased expression of α-Smooth muscle actin (αSMA) observed in many activated (myo)fibroblasts is both a reflection of the increased environmental mechanical stress (mechano-sensing) as well as the functional requirement for a contractile phenotype vital for restoring mechanical homeostasis (tissue contraction). In addition, myofibroblasts deposit an ECM rich in profibrotic mediators, including extradomain-A (ED-A) splice variant of fibronectin, periostin, tenascin-C, and latent transforming growth factor-β (TGF-β) binding protein-1 (LTBP-1), all of which facilitate the repair process [35]. Fibroblasts are able to release covalently bound latent TGF-β in the ECM, a key chemical stimulator of myofibroblast conversion [36], in a mechanical process that is enhanced in stiff ECM environment [37]. Similarly, in activated myofibroblasts Twist1 was shown to directly upregulate Prr1 expression which increases tenascin-C synthesis and reenforces Twist1 expression in a positive feedback loop [38]. Thus, myofibroblasts are prone to enter a positive mechanical feedback loop of cell contraction induced matrix stiffening and profibrotic ECM deposition/remodelling that reinforces and potentially memorises their activated state. While this multi-layered feedback loop guarantees a fast and efficient tissue repair, a persistent induction can evolve to pathological tissue fibrosis, a key characteristic of many inflammatory disorders and cancer [39].

Different mechanical cues and ECM organisations can promote distinct cellular responses. Cyclic stretching of primary human lung fibroblasts for examples inhibits myofibroblast differentiation by reduced paracrine expression of TGF-β [40]. Culturing fibroblasts on stiff and soft culture conditions have been shown to promote distinct transcriptional signatures that are able to prime cells long-term. Fibroblast expanded in stiff microenvironments maintain a profibrotic phenotype over several weeks when switched to a soft substrate. Conversely, soft substrate culture upregulates MMP-1, MMP-3 and MMP-13 production and reduces expression of fibrosis-associated genes (such as α-SMA, Col type-1, and CTGF) in skin or cardiac fibroblasts, restraining their activation when plated on a stiff culture substrate [41,42]. After myocardial infarction, the mechanical properties of the heart change regionally and over time, inducing distinct phenotypes in cardiac fibroblasts that can be recapitulated in vitro [43]. While paracrine signalling from stretched cardiomyocytes promotes fibroblast proliferation, direct fibroblast stretching induced ECM synthesis and progressive matrix stiffening lead to an upregulation of αSMA expression and a switch from type I to type III collagen production. This example emphasizes how different mechanical cues within an organ can induce distinct profibrotic phenotypes of activated fibroblasts, which need to be considered for the development of future, fibroblast-targeted therapies.

Although myocardial infarction induced fibrosis is generally thought of as irreversible, some degree of resolution can be observed after acute injury and its subsequent repair process. In addition to apoptosis, lineage tracing experiments indicate that some myofibroblasts in the heart can differentiate into a less activated and non-proliferative state [44]. This new stable cell state, referred to as matrifibrocyte, expresses an ECM gene signature rich in bone-cartilage markers such as chondroadherin and cartilage oligomeric matrix protein (Comp), reminiscent of tendon, and is anticipated to promote a mature scar, respond differently to mechanical cues and support the heart against further damages [45]. Currently, it is unclear if matrifibrocytes are also present in scar tissue from other organs. By combining ATAC-seq and RNA-seq analysis, a recent study indicates that the martrifibrocyte phenotype is maintained by changes in chromatin accessibility and gene expression [46] and harnessing this cell plasticity may be therapeutically valuable.

Within a cell these different mechanical cues are received and processed via mechanosensitive proteins at the cell membrane-cytoskeletal cortex interface. PIEZO1/1, TRPC3/6 and TRPV4 are all examples of proteins involved in the mechano-sensing that act in concert with the cytoskeleton and cell-cell and cell-ECM adhesion receptors at the cell surface (e.g., integrins and cadherins) [47]. These mechanical forces can either result in direct downstream transcriptional regulation via cytoplasmatic/nuclear localisation of YAP/TAZ, NFκB and SRF/MAL or directly affect the chromatin organisation in the nucleus [48,49,50,51]. Notably, chromatin displays rheological properties through its ability to contort under mechanical load, resulting in direct transcriptional changes [49]. Short-term application of mechanical stress of 17.5 Pa was shown to double transcription of *DHFR*, a housekeeping gene necessary for the formation of thymidine through the reduction of dihydrofolate into tetrahydrofolate [52]. In contrast, long-term stimulation resulted in heterochromatin formation via the ATP-dependent condensation pathway leading to a global reduction in gene transcription [53]. It is conceivable that both direct nuclear adaptations to mechanical stimuli as well as activity of other mechano-sensing pathways (such as YAP/TAZ) modulate fibroblast gene transcription, behaviour and fate as shown for multipotent stem cells [54,55]. In line, Roy et al., revealed that culturing of fibroblasts on micropatterned substrates that laterally confined their growth was sufficient to induce their reprogramming to a more stem cell-like state [56]. Intriguingly, this approach enabled rejuvenation of aged fibroblasts upon redifferentiation in a 3D collagen matrix, indicating a potential therapeutic application [57]. Similarly, inhibition of focal adhesion kinase (FAK), a well-established transducer of mechanical forces, reduced YAP/TAZ-ERK induced scar formation by promoting AKT-EGR1 signalling and thus a more regenerative wound repair [58].

Besides being central to mechano-sensing in multiple cell types [59], YAP/TAZ signalling was shown to promote short-term mechanical memory [50,55]. YAP/TAZ nuclear accumulation increases in culture when fibroblasts are grown on stiff matrices and knockdown of YAP or TAZ attenuates fibroblast proliferation, contraction, and matrix synthesis. Conversely, fibroblasts conditionally expressing active YAP or TAZ mutant proteins are able to maintain a high growth rate on a soft matrix and promote fibrosis when adoptively transferred to murine lungs, highlighting that YAP/TAZ activation in fibroblasts is sufficient to drive a profibrotic response in vivo [50]. Here the YAP/TAZ transcriptional target plasminogen activator inhibitor-1 (PAI-1) was shown to be central for its profibrotic function. Mechanical activation of YAP/TAZ signalling induces TGF-β independent PAI-1 expression, which inhibits plasmin-dependent proteolysis [60], a mechanism that normally disrupts YAP/TAZ signalling [61]. This feedback loop, induced by mechanical stimuli, maintains YAP/TAZ signalling and promotes TGF-β signalling, providing priming and potential memory of the mechanical microenvironment [50]. During wound healing, loss of a contractile (myofibroblast) phenotype in the late repair phase correlates with reduction in nuclear YAP/TAZ in fibroblasts [62,63], whereas persistent (irreversible) YAP/TAZ activation upon long-term culture on stiff substrates alters the differentiation potential of mesenchymal stem cells (MSCs) [55].

In addition to the YAP/TAZ pathway, SRF/Mal and NFκB signalling have been shown to be important for mechano-signalling in fibroblasts. The mechanotranduction of NFκB can be initiated via mechanical stress and is propagated by focal adhesion kinase signalling [64]. Elevated nuclear NFκB localisation has been linked to fibrosis and inflammation in multiple organs [51]. The SRF/MAL complex which is activated through Rho signalling in response to TGF-β, Wnt, integrin and cadherin signalling, induces myofibroblast differentiation during tissue repair [65,66]. Interestingly, there appears to be a crosstalk between this complex and the inner nuclear membrane protein Emerin, promoting accumulation of nuclear MAL in response to increased substrate stiffness [67]. How SRF/Mal and NFκB signalling contribute to mechanical memory is still unclear and it is possible that these pathways also affect chromatin remodelling and thus long-term changes to gene transcription. The nucleus itself is physically linked to the cytoskeleton through the Linker of Nucleoskeleton and Cytoskeleton (LINC) proteins, which themselves are anchored within the inner nuclear and outer nuclear membranes [68]. Indeed, mechanical strain induced enrichment of Emerin at the nuclear outer membrane has been shown to regulate gene silencing and chromatin compaction during the lineage commitment of epidermal stem cells [54].

Another emerging key player for mechanical memory are miRNAs, in particular miRNA-21 [69]. In MSCs, in vitro culture on stiff substrate induces a profibrotic phenotype that is maintained by miRNA-21, which itself is regulated by mechanosensitive MAL. Subsequent knockdown of miRNA-21 or priming MSCs on soft silicone substrates was sufficient to suppress a profibrotic phenotype and protects them from mechanical activation. In MSCs it has been suggested that while changes in YAP/TAZ signalling may act as a short-term memory storage, miRNA-21 is able to provide long-term memory of mechanical stimuli. In addition, miRNA-21 has also been implicated in profibrotic phenotype of cardiac fibroblasts, indicating a similar mechanism for mechanical memory which further involves direct suppression of the Smad7 signalling pathway [70].

The importance of epigenetic modifications for storing mechanical information long-term was shown in MSCs, where matrix stiffness can alter their regenerative capacity in a dose dependent manner. Analysis of histone modifications and chromatin organisation has revealed that cells rapidly respond to changes in the mechanical microenvironment and that these adaptations include distinct signatures of epigenetic modulators which become irreversible upon long-term exposure [71]. Furthermore, tissue stiffness is able to induce and maintain lung fibroblast activation by epigenetic silencing of peroxisome proliferator-activated receptor γ coactivator 1 α (*PGC-1α*) gene via H3K9 methylation through induction of histone methyltransferase G9a (EHMT2) and chromobox homolog 5 (CBX5) [72].

Thus, mechanical memory in fibroblasts appears to be determined by the type and duration (mechano-dosing) of the mechanical stimulus through induction of specific transcriptional and epigenetic signatures (Figure 2b).

## 4. Inflammatory Memory

Fibroblasts are increasingly being recognised as essential cells for the immune system [1]. They are equipped with pattern recognition receptors (PRRs) and can secrete a large range of inflammatory mediators such as cytokines and chemokines. During development, immune cells progressively populate various organs, and it is conceivable that there is an adaptive signalling crosstalk between infiltrating immune cells and various fibroblast populations in the tissue. For the adaptive immune system, immune memory is essential to mount an effective immune response to specific inflammatory stimuli, and aberrant inflammatory memory can lead to multiple chronic inflammatory diseases. Also, cells of the innate immune system exhibit ‘trained immunity’, an epigenetic memory of pathogen encounters priming the innate immune system for repeat exposure. Thus, it is not surprising that fibroblasts, as mediators of inflammation, are capable of memorising inflammatory insults as well. Indeed, fibroblasts are emerging as key players in several chronic inflammatory conditions, such as rheumatoid arthritis [73], Lyme arthritis [74], scleroderma [75], and atopic dermatitis [76]. Therefore, it is crucial to understand how fibroblasts respond to inflammation and how fibroblast immune memory can prolong inflammatory stress following the initial exposure.

During tissue damage-induced inflammation, ECM remodelling can release damage associated molecular patterns (DAMPs), such as fibronectin fragments, from unfolding or enzymatic digestion [77], while wound infiltrating pathogens release pathogen associated molecular patterns (PAMPs) such as bacterial lipopolysaccharides (LPS). Fibroblasts can sense DAMPs and PAMPs through a repertoire of toll-like receptors (TLRs) with substantial heterogeneity in sensitivity [78] and subsequently release cytokines, remodel ECM and attract immune cells. Tolerance to repeated TLR signalling is important for preventing damage from an excessive inflammatory response. LPS is an agonist of TLR4, which is expressed on the surface of multiple tissue fibroblasts, however the ability to memorize TLR4 stimulation varies among different fibroblast populations. Repeated LPS stimulation causes loss of activating histone marks in anti-viral genes in dermal fibroblasts and fibroblast-like synoviocytes (FLS), but not in gingival and neonatal foreskin fibroblasts [79].

In addition to TLRs, fibroblasts can sense inflammatory signals through cytokine receptors, which can lead to long-term changes in fibroblast behaviour. For instance, prolonged TNFα signalling causes chromatin remodelling in FLS, accompanied by an increase in nuclear localisation of NFκB [80,81,82]. This is not only observed in FLS, but also in fibroblasts from chronically inflamed gum and skin. Mechanistically, a three-day exposure of fibroblasts to TNFα results in an increased response to interferon stimulation for several days after TNFα removal, which was characterised by decreased histone abundance and increased histone acetylation and STAT-1 signalling [80]. Thus, in fibroblasts, a memory of the TNFα stimulation is maintained through a combination of chromatin remodelling and sustained increased expression of pro-inflammatory cell signalling pathways.

Besides TNFα, several interleukins (IL) have been implicated in the inflammatory memory of fibroblasts. IL-8 is important for attracting immune cells and promoting phagocytosis and angiogenesis. In normal wounds, IL-8 levels are low, but burn wounds often have slow healing areas with elevated IL-8 expression. Exposing dermal fibroblasts to elevated levels of IL-8 in vitro was shown to inhibit fibroblast long-term contraction [82], and it is hypothesised that this memory effect could alter ECM deposition/remodelling and influence epithelial cell migration during wound repair [83]. Viral infections induce secretion of IL-13, which can prime fibroblasts for conversion to tertiary lymphoid structures [84]. The IL-13 response in fibroblasts is modulated by miRNA-135b, which is downregulated through DNA hypermethylation in systemic sclerosis [85]. In addition, miRNAs that are associated with inflammatory memory in immune cells [86] have also been shown to regulate various fibroblast functions [87,88,89]; whether they play a role in fibroblast inflammatory memory is unclear.

The inflammatory fibroblast phenotype is also influenced by the metabolic state. Increased expression of hexokinase 2 (HK2), a key enzyme of glucose metabolism, results in an invasive and migratory cellular state in rheumatoid arthritis associated FLS [90]. Knockdown of HK2 in mice was shown to reduce arthritis severity, bone, and cartilage damage, suggesting a direct link between immune response and metabolic deregulation in fibroblasts [90]. Pericytes are a specialised fibroblast subpopulation that reside on blood vessels at the interface between the blood and connective tissue and are involved in controlling immune cell migration through expression of cytokines and adhesion molecules. It has been shown that retinal pericytes retain a memory of high glucose and respond with a sustained inflammatory phenotype, even when returned to normal glucose conditions [91]. Currently, the underlying mechanism remains unclear; however, epigenetic profiling of vascular smooth muscle cells from diabetic mice suggests that it could be mediated by removal of repressive histone modifications at proinflammatory promoters [92]. Similarly, whether pericytes from other tissues (e.g., skin) are capable of maintaining an inflammatory memory of high glucose exposure warrants further investigation.

In summary, fibroblasts have been demonstrated to function as innate immune cells and maintain a memory, or trained immunity, to repeated inflammatory stimuli. This memory is maintained through a variety of mechanisms including DNA methylation, histone modifications and changes in histone abundance, miRNAs, sustained transcription factor and signalling pathway activities, and their metabolic state (Figure 2c). Dysregulation of fibroblast inflammatory memory can be a direct result of disease but also act to maintain and prolong chronic diseases and tissue damage.

## 5. Metabolic Memory

Cellular metabolic state is an important regulator of fibroblast behaviour in development, homeostasis, wound healing, and disease. The metabolic programme (the balance in a cell between metabolic pathways such as glycolysis, oxidative phosphorylation, and lipolysis) regulates energy and metabolite intensive activities such as ECM production and reorganisation, myofibroblast contraction, migration, and proliferation [93,94]. While most fibroblasts are proliferative during development, in homeostasis adult fibroblasts generally exist in a quiescent state with a distinct metabolic programme [34], which is actively maintained by repressing proliferation and the transition into senescence or terminal differentiation [95,96]. Notably, quiescent fibroblasts remain highly metabolically active and increase expression of ECM proteins such as collagen I and III, which is partly controlled by miRNAs including miRNA-29 [97,98].

In disease the balance of oxidative phosphorylation, aerobic glycolysis and fatty acid oxidation can become dysregulated, as observed in skin fibrosis with an increase in glycolysis and decrease in fatty acid oxidation [93]. Fibroblasts can sense intracellular and extracellular metabolic changes in their microenvironment through metabolic sensors such as C-terminal binding protein 1 (CtBP1) and respond with an altered metabolic programme and cell behaviour [99]. Intriguingly, these responses can persist even after the metabolic stimulus has subsided, suggesting that fibroblasts are capable of memorising specific changes in metabolism. The oncogenic metabolic phenotype of cancer associated fibroblasts (CAFs), for example, was recently shown to be maintained through a reduction of DNA and histone methylation caused by a nicotinamide N-methyltransferase (NNMT) induced depletion of S-adenosyl methionine (SAM), a universal methyl donor [100].

In the lungs, arteries and heart, changes in blood pressure and the availability of oxygen to fibroblasts can signal tissue damage or disease. Hypoxia-induced pulmonary hypertension, for example, causes adventitial fibroblasts to switch to glycolytic metabolism with corresponding increase in NADH. The high NADH levels are sensed by fibroblasts though CtBP1, which promotes a pro-inflammatory and proliferative cellular state. When these adventitial fibroblasts are returned to normoxia culture conditions, they maintain a persistent glycolytic programme characterised by increased proliferation and inflammatory signalling which can be reversed by pharmacologically reducing NADH or silencing CtBP1 [99]. In addition, the persistent hypoxia-induced profibrotic changes in adventitial fibroblasts have been recently linked to specific alterations in mitochondrial metabolism in pulmonary hypertension conditions leading to a metabolic pyruvate to lactate shift and increased mitochondrial superoxide production [101]. Also cardiac fibroblasts are able to maintain a metabolic memory of hypoxia, promoting a persistent profibrotic environment with increased proliferation, fibroblast activation and excessive collagen and cytokine secretion [102,103]. These pro-fibrotic changes are associated with global DNA hypermethylation and increased expression of the DNA methyltransferase (DNMT) enzymes DNMT1 and DNMT3B, which is controlled by hypoxia-inducible factor (HIF)-1α. DNMTs depletion or inhibition significantly reduces collagen deposition, αSMA expression and response to profibrotic cytokines in cardiac fibroblasts [103]. Beside promoting the expression of fibrogenic cytokines, like TGF-β1 and CTGF, HIF-1α, it was shown to increase the expression of pyruvate dehydrogenase kinase (PDK) in profibrotic cardiac fibroblasts. PDK inhibition reverses the mitochondrial-metabolic phenotype and decreases fibroblast proliferation and collagen production in vitro [102]. Mechanistically, epigenetic suppression of the mitochondrial gene superoxide dismutase 2 (*SOD2*) by DNA methylation causes decreased mitochondrial hydrogen peroxide signalling and a metabolic shift with increased uncoupled glycolysis, which is maintained by the DNMT-HIF-1α-PDK feedback loop and global DNA hypermethylation. Thus, metabolic memory of hypoxia in fibroblasts is mediated by a combination of sustained transcriptional repressor activities (CtBP1), epigenetic changes (DNA methylation) and altered mitochondrial metabolism.

Fibroblasts have also been shown to sense and memorise increased exposure to specific metabolites such as glucose. Diabetes is a disease with increased risk of hyperglycaemia and is associated with metabolic dysregulation such as reduced perfusion and oxygenation, and increased catabolism through hormone signalling [104]. Analysis of dermal fibroblasts from diabetic patients shows altered gene expression when exposed to high glucose conditions in vitro, compared to fibroblasts from non-diabetic donors [105]. In fibroblasts, hyperglycaemia results in a pattern of DNA methylation surrounding genes associated with wound repair, angiogenesis, and ECM assembly, which persists for multiple passages in vitro in normoglycemic conditions. It has been suggested that this metabolic priming of dermal fibroblasts may contribute to the impaired wound healing observed in diabetic patients [106]. Indeed, cultured fibroblasts from type 2 diabetes patients show decreased sensitivity to TNFα stimulation compared to healthy donors, which is probably driven by epigenetic modifications [107]. Whether these changes in fibroblast behaviour and hyperglycaemic memory are limited to dermal fibroblasts, or extend to fibroblasts in other organs, warrants further investigation.

The metabolic state of fibroblasts can be influenced by several paracrine and autocrine signals from neighbouring cells within the microenvironment. A hallmark of cancer is a dysregulated metabolism in which cancer cells undergo metabolic reprogramming to aerobic glycolysis, providing metabolites for cell division. This metabolic switch is known as the “Warburg effect” and has also been observed in activated T cells and fibroblasts [108,109,110]. Moreover, cancer cells can induce aerobic glycolysis in neighbouring fibroblasts to provide an environment rich in metabolites needed for anabolism, in a process termed the “Reverse Warburg effect”. In breast cancer autocrine and paracrine TGF-β signalling have been shown to induce downregulation of the membrane protein caveolin-1 (Cav-1) in CAFs. Loss of Cav-1 causes CAF metabolic reprogramming to aerobic glycolysis, mitochondrial dysfunction and increased autophagy/mitophagy, which propagates to neighbouring CAFs and promotes the anabolic growth of adjacent cancer cells [111]. Notably, in rheumatoid arthritis, T helper cells have been shown to reprogramme fibroblasts to a glycolytic phenotype [112], suggesting that fibroblast metabolic reprogramming is a common feature in cancer and inflammatory diseases. Similarly, this phenotype can be further induced and maintained by autocrine TGF-β signalling and is able to spread to adjacent fibroblasts through paracrine signalling [111]. An autocrine TGF-β signalling loop, as a form of memory, is also observed in kidney fibrosis, in which myofibroblast-induced tension causes the release of TGF-β1, prolonging a contractile and glycolytic phenotype [113].

In conclusion, metabolic memory in fibroblasts arises from local and systemic perturbations to their environment such as hypoxia and hyperglycaemia but can also be induced by neighbouring cells in cancer and inflammatory diseases. This memory is maintained through several mechanisms, including sustained autocrine/paracrine signalling of cytokines, sustained transcription factor activity, miRNAs and epigenetic modifications to DNA and histone proteins (Figure 2d).

## 6. Conclusions and Outlook

Fibroblasts, present in almost every organ, are increasingly understood to have a diverse range of functions, contributing to tissue homeostasis, wound healing, defence from pathogens and damage, and metabolism. Their memory of the systemic and local (micro) environment is essential for tissue development and maintenance, but can become dysregulated in disease, contributing to fibrosis, chronic inflammation, poor wound healing, and cancer. Here, we have discussed four key areas of fibroblast memory, but there may be more yet to be uncovered. Each type of memory is maintained by a distinct combination of epigenetic modifications, DNA binding factors, cytoplasmatic components, cell-cell/ECM interactions, and signalling factors, all of which depend on the sensed stimulus and enable fibroblasts to store the information of their microenvironment for short or long periods of time. While positional memory, established during early development, appears to be very stable, involving specific epigenetic changes and HOX gene signatures, others, such as mechanical or inflammatory memory, are more transient and able to rapidly adapt to environmental changes. Fibroblast mechanical memory, for example, has been proposed to emerge through a stepwise process of transcriptional reinforcement of cytoskeletal signals, expression of memory-regulating factors, and reduction in epigenetic plasticity, which inhibits further mechanical adaptations [114]. How short- and long-term (potentially pathological) inflammatory and metabolic memory become established is far less understood. To reduce the risk of chronic inflammation, some fibroblast populations could have a memory of repeated inflammatory insults and subsequently become tolerised, while others may develop an inflammatory phenotype that fails to resolve. In homoeostasis, fibroblasts are generally quiescent but become highly metabolically active during tissue repair which can persist even when the stimulus is removed and eventually becomes irreversible in disease conditions such as fibrosis.

Therefore, modulating distinct fibroblast memories that may promote pathological fibroblast behaviours presents a promising therapeutic target. Here, an important question is how different types of memory are connected and influence each other in homeostasis and disease. During development and tissue repair, mechanical and metabolic changes are tightly linked, as metabolic adaptions are essential for enhanced fibroblast functions, including contraction and ECM deposition/remodelling. Indeed, besides a stiff microenvironment, a persistent dysregulated metabolism towards increased glycolysis has been shown to maintain fibroblasts in a profibrotic state in fibrosis and cancer [93]. Likewise, fibroblast metabolic and inflammatory memory seem to be closely associated. While metabolic memory in FLS and retinal pericytes is able to induce a persistent proinflammatory phenotype, activated T cells have been shown to reprogram fibroblast metabolism in inflammatory rheumatoid arthritis [90,112]. The way in which positional memory confers site-specific fibroblast behaviours and links to other environmental adaptations is largely unknown. A recent study comparing healthy and arthritic synovial fibroblasts from different anatomical sites suggests that the local identity and microenvironment of stromal cells predisposes to the development of positional disease patterns [10]. The prominent chemotactic and ECM degradation phenotype observed in arthritic synovial fibroblasts from the hand may explain the more aggressive and destructive disease progression in hands compared to other joints. Similarly, keloid scars, caused by an abnormal wound healing response, occur predominantly in posterior skin regions of the ears, face and upper torso. This positional predisposition has been associated with the expression of distinct HOX gene signatures but also differences in skin tension [115] and ECM deposition/remodelling [116]. Conversely, fibroblasts from the oral cavity have an increased regenerative potential and understanding the molecular and environmental features determining their functional diversity may help to develop fibroblast-targeted anti-fibrotic therapies. Furthermore, a CAF subset in axillary lymph nodes has been shown to promote breast cancer metastasis formation, emphasising how fibroblasts residing in distinct anatomical locations can influence disease progressions across different body sites [117].

The prominent role of epigenetics for maintaining fibroblast memory implies that fibroblasts could be targeted by epigenetic modulators, such as histone deacetylase inhibitors (HDACi) in disease. HDAC2 was found to be increased in normal and keloid scars, [118] and it was hypothesised that HDACi treatment may decrease skin fibrosis, as proposed in other organs including the kidneys [119,120] and heart [121]. HDAC inhibition may also alleviate metabolic memory by reducing renal fibrosis associated with diabetes [122] and reversing the fibroblast glycolytic phenotype in pulmonary hypertension [123]. While HDAC inhibition has been shown to reduce inflammation and may be used to target fibroblasts in chronic inflammatory diseases, this effect seems to be due to a combination of modulating acetylation of histones and non-histone signalling proteins [124,125]. Furthermore, disrupting the sensing of the mechanical microenvironment may prove effective in treating fibrosis. Pharmacological inhibition of the focal adhesion kinase or YAP/TAZ signalling pathway was recently shown to significantly reduce scar formation in the skin [58,126,127]. Currently, the concept is emerging that any disease with a prominent stromal component harbours positionally imprinted ‘risk’ signatures (memory), which develops into a positional disease pattern that may differ in disease progression and therapeutic response. Comparison of dermal fibroblasts extracted from different anatomical locations revealed an astonishing variability in their ability to be reprogrammed to an induced pluripotent state, pointing to the importance of positional heterogeneity for the development of future therapeutic applications involving induced pluripotent stem cell technology [128]. Recent advances in generating a cross-tissue atlas of gene expression in fibroblasts, combining single-cell RNA datasets across different tissues, species, and diseases, are now allowing the definition of the universal (pan) fibroblast phenotypes and tissue- and disease-specific subsets or cellular states in multiple organs [5]. These are exciting new tools to dissect the molecular mechanisms and interplay of different functional adaptations/memories in fibroblasts, which will potentially pave the way to revolutionize current disease diagnosis, patient stratification and therapy development.

## Figures and Tables

**Figure 1 cells-10-02840-f001:**
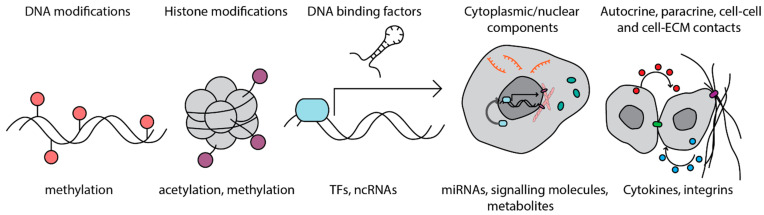
Mechanisms of biological memory: memory in cells is maintained on several levels, including direct modifications to DNA, changes to histone proteins to alter DNA accessibility, sustained expression of DNA binding factors and nuclear components, increased abundance or activity of cytoplasmic components, such as miRNAs, metabolites and signalling molecules, and through cell surface receptors sensing autocrine and paracrine signals and direct cell-cell and cell-ECM contacts. TF, transcription factor.

**Figure 2 cells-10-02840-f002:**
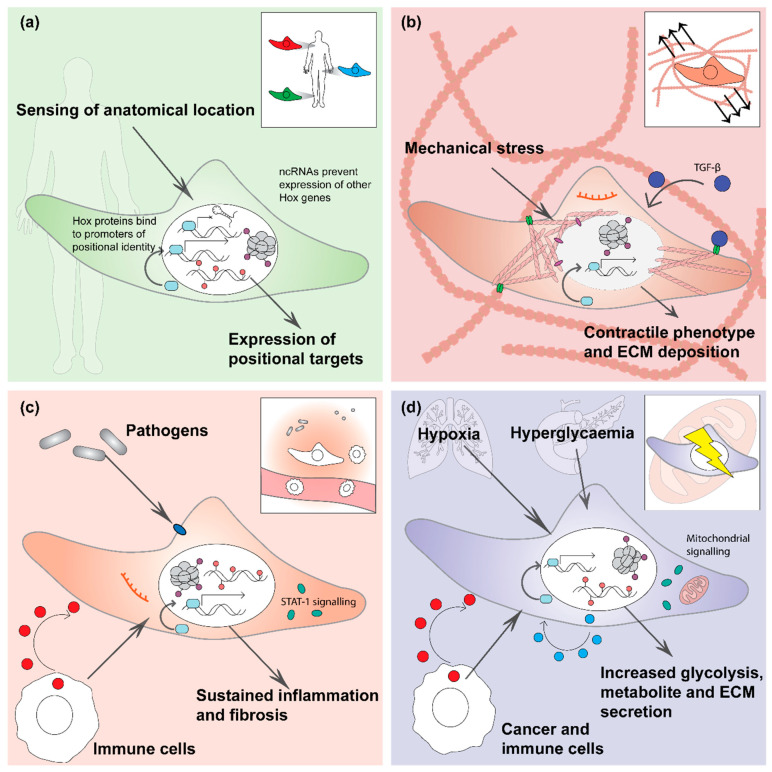
Types of fibroblast memory: (**a**) positional memory is established during development and is largely driven by Hox gene expression. The expression of region specific Hox genes is maintained by autoregulatory loops, modifications to DNA and histones and by ncRNA inhibiting expression of specific Hox genes; (**b**) mechanical memory: fibroblasts are linked to the external mechanical environment of the ECM by integrins and other adhesion receptors. Mechanical strain transmitted via the cytoskeleton can directly influence chromatin accessibility and gene transcription. Beside chromatin modifications, mechanical memory is maintained through expression of miRNAs (e.g., miR-21), sustained TF activity such as NFκB and YAP/TAZ, and positive feedback loops such as contraction induced release of latent TGF-β from the ECM; (**c**) inflammatory memory: fibroblasts sense the inflammatory state through pattern recognition and cytokine receptors. Memory is maintained through modifications to DNA and histones, sustained TF and signalling pathway activity, such as the JAK/STAT pathway, and possibly through miRNAs; (**d**) metabolic memory in fibroblasts is maintained by modifications to DNA and histones, sustained TF activity, an autocrine TGF-β feedback loop and positive feedback involving altered mitochondrial signalling and cellular metabolite levels.

## Data Availability

Not applicable.

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
