# Peer review of "Fibroblast Memory in Development, Homeostasis and Disease"

_cells, 2021, doi:10.3390/cells10112840_

Round 1
Reviewer 1 Report
The present review discusses the concept of memory in fibroblasts highlighting positional, mechanical, inflammatory, and metabolic memory. The authors describe the underlying pathways that are implicated in maintaining the adaptations in these four areas. The review is well written and provides a unique perspective on fibroblast adaptations in both normal and pathological contexts. This review is timely as there is a growing interest in identification and targeting of fibroblast sub-populations as a result of new single cell RNA sequencing studies, as well as in metabolic activity in fibroblasts and epigenetic processes as a target for anti-fibrotic therapy. The following came up during review and are presented for consideration.
- The authors nicely highlight epigenetic mechanisms that underlie fibroblast memory. A section on the degree to which targeting these epigenetic processes is a viable therapeutic option would strengthen the review. For example, certain HDAC inhibitors have been shown to have anti-fibrotic effects in settings of pathologic remodeling. What are the implications on mechanical, inflammatory, and metabolic memory?
- With respect to mechanical memory, it would be of interest to include the transition that fibroblasts undergo in the post-infarct myocardium. Specifically, following initial wound healing response, the cells transform into a new phenotype to maintain the mature scar. These cells are termed matrifibrocytes and likely are responding to different mechanical cues than in the active remodeling phase.
- Phenotypic shifts have been shown in vivo either towards a more pathological phenotype or less Fibrogenic phenotype. The review nicely covers the shift toward fibrosis. The authors should consider also discussing the cellular memory following reversal of pathology as has been shown either following removal of the pro-fibrotic stimulus or following a therapeutic intervention.
Author Response
Comments and Suggestions for Authors
The present review discusses the concept of memory in fibroblasts highlighting positional, mechanical, inflammatory, and metabolic memory. The authors describe the underlying pathways that are implicated in maintaining the adaptations in these four areas. The review is well written and provides a unique perspective on fibroblast adaptations in both normal and pathological contexts. This review is timely as there is a growing interest in identification and targeting of fibroblast sub-populations as a result of new single cell RNA sequencing studies, as well as in metabolic activity in fibroblasts and epigenetic processes as a target for anti-fibrotic therapy. The following came up during review and are presented for consideration.
Response: We want to thank the reviewer for the positive evaluation of our review manuscript.
Point 1. The authors nicely highlight epigenetic mechanisms that underlie fibroblast memory. A section on the degree to which targeting these epigenetic processes is a viable therapeutic option would strengthen the review. For example, certain HDAC inhibitors have been shown to have anti-fibrotic effects in settings of pathologic remodeling. What are the implications on mechanical, inflammatory, and metabolic memory?
Response: We want to thank the reviewer for this excellent suggestion and have highlighted more clearly the therapeutic links throughout the manuscript and added an extended section in the conclusion part.
Point 2. With respect to mechanical memory, it would be of interest to include the transition that fibroblasts undergo in the post-infarct myocardium. Specifically, following initial wound healing response, the cells transform into a new phenotype to maintain the mature scar. These cells are termed matrifibrocytes and likely are responding to different mechanical cues than in the active remodeling phase.
Response: We have now included this information in the mechanical memory section.
Point 3. Phenotypic shifts have been shown in vivo either towards a more pathological phenotype or less Fibrogenic phenotype. The review nicely covers the shift toward fibrosis. The authors should consider also discussing the cellular memory following reversal of pathology as has been shown either following removal of the pro-fibrotic stimulus or following a therapeutic intervention.
Response: We feel that we have indicated the changes in fibroblast memory upon reversal of the pathological stimulus in the respective sections at multiple occasions. For example, we describe the changes when fibroblasts are switched from stiff to soft culture environment (loss of mechanical stimulus) and from high glucose/hypoxia to normal culture conditions (loss of metabolic stimulus) as well as how they respond upon TNFα stimulation withdrawal (loss of inflammatory stimulus) or transplantation (loss of anatomical location). The addition of a section about the therapeutic targeting of fibroblast memory (Point 1) is now further strengthening this point.
Reviewer 2 Report
This is a very well written and timely review on the topic of fibroblast memory. It will be of interest to all those who wish to be updated to the ever-changing and complex field of fibroblast biology. I have no concerns. Minor and optional suggestions are as follows:
(1) The title speaks of fibroblast memory in development, homeostasis and disease, yet the manuscript is structured so as to describe different modes of memory. The authors might outline (coming full circle) similarities and difference between developmental and disease-related fibroblast observations.
(2) The authors might wish to speculate as to what possible evolutionary advantage (if any) does the persistence of location-specific characteristics confer to the various fibroblasts. Could it be a feature that might facilitate better repair/regeneration post-injury?
Author Response
Comments and Suggestions for Authors
This is a very well written and timely review on the topic of fibroblast memory. It will be of interest to all those who wish to be updated to the ever-changing and complex field of fibroblast biology. I have no concerns. Minor and optional suggestions are as follows:
Response: We want to thank the reviewer for the positive evaluation of our review manuscript.
Point 1. The title speaks of fibroblast memory in development, homeostasis and disease, yet the manuscript is structured so as to describe different modes of memory. The authors might outline (coming full circle) similarities and difference between developmental and disease-related fibroblast observations.
Response: We want to thank the reviewer for this suggestion and have strengthen the potential implications of fibroblast memory for development in the mechanical, metabolic and inflammatory memory section.
Point 2. The authors might wish to speculate as to what possible evolutionary advantage (if any) does the persistence of location-specific characteristics confer to the various fibroblasts. Could it be a feature that might facilitate better repair/regeneration post-injury?
Response: This is indeed an intriguing question and we have included a speculation at the beginning of the positional memory section.